# The role of mental health in the relationship between nursing care satisfaction with nurse-patient relational care in Chinese emergency department nursing

Hui Huang[1], Jing Cui[2], Hua Zhang[3], Yuhui Gu[4], Haosheng Ni[5], Ya Meng[6]*

1 Department of Infusion Room for Adults, Affiliated Hospital of Nantong University, Nantong University, Nantong, China, 2 Department of Rehabilitation, Hai'an Traditional Chinese Medicine Hospital, Hai'an, China, 3 Department of Urinary Surgery, Affiliated Hospital of Nantong University, Nantong University, Nantong, China, 4 Department of Endoscopic Center, Affiliated Hospital of Nantong University, Nantong University, Nantong, China, 5 Department of Otolaryngology, Affiliated Hospital of Nantong University, Nantong University, Nantong, China, 6 Department of Outpatient Injection, Affiliated Hospital of Nantong University, Nantong University, Nantong, China

* mengya202312@163.com, rapagee1@gmail.com

**Data Availability Statement:** All relevant data are within the manuscript.

**Funding:** This study was funded by Nantong Science and Technology Project, China (Grant

## Abstract

### Background

The relationship between a nurse and a patient is a key part of nursing that can impact how happy the patient is with the care they receive. It appears that the nurse's mental health can also affect this connection. However, there is little research on this topic. So, the aim of the present study was to determine the correlation of nurse's mental health with nurse–patient relational care and nursing care satisfaction.

### Methods

A total of 532 nurses and 532 patients from 13 Level-III hospitals of Hubei province (China) completed a China Mental Health Survey, general information questionnaire, the Nursing Care Satisfaction Scale, and Relational Care Scale.

### Results

Age, nurse working years, and night shift last month were correlated with mental health score ($r = -0.142$, $r = -0.150$, $r = 0.164$, $p < 0.05$). Nurse's mental health was correlated with relational care score and nursing care satisfaction score ($r = -0.177$, $r = -0.325$, $p < 0.05$). Also, relational care score, patients age and gender were correlated with nursing care satisfaction score ($r = 0.584$ and $r = 0.143$, $x^2 = 11.636$, $p < 0.05$). Descriptive information of nurses had a direct impact on nurses' mental health (direct effect = 0.612, 0.419–0.713). Nurses' mental health had a direct effect on relational care score (direct effect = 0.493, 0.298–0.428) and an indirect effect on nursing care satisfaction score (indirect effect = 0.051, 0.032–0.074). Relational care score and patient's descriptive information had also a

number: MS22022113). The funders had no role in study design, data collection and analysis, decision to publish, or preparation of the manuscript.

**Competing interests:** The authors have declared that no competing interests exist.

direct effect on nursing care satisfaction score (direct effect = 0.232, 0.057–0.172 and 0.057, 0.347–0.493).

## Conclusion

This study showed that the better the mental health of nurses, the more patients feel satisfied with nursing services.

## Introduction

Every patient has the right to feel satisfied with the services provided by the hospital. For this reason, hospitals and many healthcare centers strive to fulfill patients' needs through the provision of quality services such as shortening wait times and preventing unnecessary delays that can be harmful, ensuring that the quality of care is consistent, regardless of a person's gender, race, where they live, or how much money they have, offering a complete set of health services for all stages of life, and using resources effectively to get the most benefit and not wasting them [1].

Patient satisfaction with the services provided by the hospital has various benefits. For example, it less likely that patient will sue for malpractice. It also encourages patients to be more involved in their treatment, helps healthcare centers do better financially in a competitive market, and increases the chances of improving patients' overall health [2].

In China, there has been more focus on patient-centered care in recent years. So that, the Chinese government started a big program in 2015 to make healthcare better across the country, with goals to improve how patients feel about their care and their experiences [3]. In this system, healthcare policymakers and providers have realized the significance of patient satisfaction and have begun to view it as an indicator of care quality [4]. To achieve this goal, it is crucial to examine the diverse aspects of structural, medical, nursing, and support services. These factors contribute to the overall experience of patients.

In hospitals, emergency departments play a crucial role in shaping a patient's first impression as they enter the healthcare system [5]. A visit to the emergency department is usually the first time a patient interacts with a hospital system. Thus, this is a special opportunity to make a positive first impression.

However, the challenges in creating a pleasant experience should not be underestimated. Many patients consider their visit to the emergency department as the worst day of their lives. This already negative view makes it harder to provide a good experience [6]. Moreover, patient satisfaction in the emergency department depends on knowing how long they'll wait, the conditions of the section, and how well it's organized [7].

Also, the quality of interactions with emergency staff, like nurses, also impacts patient satisfaction [8]. Some studies in emergency departments showed that patients who saw a specific healthcare provider and younger patients had more negative communication experiences than other patients [9, 10].

Nurses with good mental health show better communication skills. They listen carefully, understand patients' concerns, and offer caring responses [11, 12]. Although many studies have been done to improve patient satisfaction, only a few have focused on communication between patients and nurses. In this study, it is assumed that the nurse's mental health affects the relationship between nursing care satisfaction scale and relational care. Therefore, the aim

of the present study was to determine the mental health of nurses and its relationship with nurse–patient relational care and nursing care satisfaction.

## Method

### Study design and participants

We carried out a cross-sectional, hospital-based, multicenter study in Hubei province, China, from November 10 to December 30, 2023. According to China's hospital grading system, hospitals were categorized into three levels from Grade III to Grade I, with Grade III hospitals representing advanced medical and nursing capabilities. To equalize the conditions, only Grade III hospitals were selected. Since three cities lacked Level-III hospitals, a total of 13 Level-III hospitals were included. The nurses who met the inclusion criteria were as follows: (1) nurses working in an emergency department unit; (2) with at least one year of working experience; (3) willing to provide consent for participation in the study. On the other hand, the nurses who did not meet the inclusion criteria were excluded based on the following criteria: (1) nurses who had taken a leave for more than six months in the previous year due to different reasons; (2) nurses who were absent from work (either on leave or vacation) during the administration of the questionnaire; (3) nurses who were not directly involved in patient care, such as the head of the nursing department and head nurse. The patients who met the inclusion criteria were as follows: (1) patients who needed emergency department services; (2) patients who were under the care of one of the selected nurses in our study; (3) patients who expressed their willingness to participate in the study by giving consent. On the other hand, the exclusion criteria included: (1) patients with cognitive impairment or those unable to complete the survey; (2) patients who had already been discharged. This study was performed in line with the principles of the Declaration of Helsinki. Approval was granted by the medical research ethics committee of affiliated hospital of Nantong university (No.20231030-42). Also, written consent was received from the participants to participate in the study.

The minimum sample size was determined by estimating the sample size according to the WHO recommendations for epidemiological studies [13]. The confidence interval (CI) was set at 95%, the standard deviation (SD) was 0.5, and the margin of error was 0.5. Additionally, a 10% contingency was added to account for non-response. Finally, the minimum sample size was 430. We invited all the nurses who had been employed in the emergency department of these 13 Level-III hospitals to participate in the study. At first, questionnaires related to nurses were distributed (November 10 to 30, 2023). After identifying the nurses who filled the questionnaires correctly, a questionnaire was given to one of the patients who received services from these nurses (December 1 to 30, 2023.)

For the purposes of modeling, values were assigned to variables including nurse and patient age (18–64 years old), nurse and patient gender (female and male), nurse working years ($\geq$1 year), marital status of nurse and patient (married, never married, and divorced/deceased), night shift last month, weekly hours of working, nurse mental health, relational care, nursing care satisfaction, and annual income (Yuan) of patients,

### Demographic information

In the present study, demographic information of nurses was including gender, age, years of working, night shift last month, marital status, and weekly hours of working were recorded. The demographic information of patients was including gender, age, annual income, and marital status.

### Nursing care satisfaction scale

In this study, the scale designed by the Nursing Care Quality Control Committee of Houston Health Care System was used [14]. The Chinese version of this questionnaire has already been used in other studies [15]. This particular scale comprises of 20 items that was assigned a score ranging from 1 to 6 points, with options such as "never," "rarely," "sometimes," "often," "most of the time," and "always." The total score on the scale amounted to 120, with higher scores indicating greater patient satisfaction with the nurses' care. The reliability of this scale in the study was assessed using Cronbach's alpha coefficient, which yielded a value of 0.92, indicating good reliability.

### Relational care scale

The scale developed by Ray and Turkel (2001), was used to measure nurse-patient relationship [16]. The patient version comprises 15 items categorized into three dimensions: work trust, ethics, and care. Each item is rated on a scale of 1 to 5, ranging from "strongly disagree" to "strongly agree". The total score possible is 75 points, with a higher score indicating a stronger nurse-patient relationship. The Chinese version of this questionnaire has already been used in other studies [15]. In this study, the Cronbach's alpha coefficient for the overall scale was 0.83.

### Mental health

The GHQ-12 assesses the mental health status of nurses through 12 self-assessment items. Each item presents four options (A, B, C, and D), and utilizes a bimodal scoring method (0-0-1-1). The scoring method was as follows selecting A or B results in a score of 0, while choosing C or D leads to a score of 1. A cumulative score of $\geq 4$ was considered a positive mental health screening rate, indicating conditions such as anxiety, depression, or insomnia [17]. A higher score indicates a more severe mental health condition. The Chinese versions of GHQ-12 for professional groups, healthcare workers, and the general population demonstrated high internal consistency [18, 19] featured good reliability and validity. In this study, the Cronbach's alpha coefficient of this scale was 0.89.

### Statistical analysis

Demographic data, mental health, and nursing care satisfaction scale and relational care scale were summarized using descriptive statistics and analyzed through the correlation matrix, *chi-square* ($\chi^2$) test, and nonparametric tests in SPSS version 26.0 (IBM Corp., Armonk, NY, USA). Proportions were utilized for qualitative and ordinal data, while means and standard deviations (SD) were used for quantitative data. The structural equation modeling was carried out with AMOS 26.0 version. The fitness of the model was assessed using fitness indices such as goodness-of-fit index (GFI), the normal fit index (NFI), comparative fit index (CFI), adjusted goodness-of-fit index (AGFI), root mean square error of approximation (RMSEA), and Tacker-Lewis index (TLI). The GFI and the AGFI should be > 0.95 and > 0.90 [20]. CFI should be > 0.96 [21] or > 0.90 [20]. NFI should be > 0.90 [20] or > 0.95 [22]. RMSEA should be < 0.05 [23].

## Results

In total, 582 questionnaires were filled by nurses. However, 532 questionnaires were approved and incompletely filled questionnaires were not included in the statistical analysis. According to the selection of one patient from each nurse who provided services to her, a total of 532 patients' questionnaires were included in the statistical analysis. The characteristics of the

**Table 1. The characteristics of the participants.**

|  |  | Nurse | Patient |
|---|---|---|---|
| Age (year) |  | 26.69 ± 4.34 years | 41.921 ± 17.49 |
| Gender (%) | Female | 69% | 53% |
|  | Male | 31% | 47% |
| Marital status (%) | married | 75% | 71% |
|  | never married | 16% | 11% |
|  | divorced/deceased | 9% | 18% |
| Working years (year) |  | 7.00 ± 3.815 |  |
| Weekly hours of working (hour) |  | 41.185 ± 4.314 |  |
| Night shift last month (day) |  | 4.822 ± 4.542 |  |
| Mental health |  | 2.52 ± 1.529 |  |
| Annual income (Yuan) |  |  | 333,765 ± 311,476.5 |
| Satisfaction with nursing care |  |  | 53.9 ± 34.793 |
| Relational care |  |  | 38.902 ± 18.825 |

participants are shown in detail in Table 1. 69% of nurses and 53% of patients were female. 75% of nurses and 71% of patients were married. The mean of working years, night shift per last month, and weekly hours of nurses were 7.00 ± 3.815 years, 4.822 ± 4.542 night, and 41.185 ± 4.314 hours, respectively. Also, the average score of nurses' mental health was 2.52 ± 1.529 and the positive rate of mental health status was 26.19%.

The mean annual income of patients, patients' satisfaction with nursing care and relational care were 333,765 ± 311,476.5 Yuan, 53.9 ± 34.793 and 38.902 ± 18.825, respectively. No significant difference was observed between female and male nurses and female and male patients in the research variables ($p > 0.05$).

Pearson correlation and $\chi^2$ tests were used to explore the correlation between variables (Tables 2 and 3). Nurse age was correlated with nurse working years ($r = 0.841$, $p < 0.05$), night shift last month ($r = -0.226$, $p < 0.05$) and mental health score ($r = -0.142$, $p < 0.05$). Nurse working years was correlated with mental health score ($r = -0.150$, $p < 0.05$). Night shift last month was correlated with weekly hours of working ($r = 0.164$, $p < 0.05$) and mental health ($r = -0.275$, $p < 0.05$). Nurse mental health was correlated with relational care score ($r =$

**Table 2. Correlation between study variables.**

|  | 1 | 3 | 4 | 5 | 6 | 7 | 8 | 9 | 10 | 12 | 13 |
|---|---|---|---|---|---|---|---|---|---|---|---|
| 1 | 1 | 0.841* | 0.092 | -0.226* | -0.112 | -0.142* | 0.091 | 0.012 |  |  |  |
| 3 | 0.841* | 1 | 0.101 | -0.041 | -0.026 | -0.150* | 0.059 | 0.078 |  |  |  |
| 4 | 0.092 | 0.101 | 1 | -0.081 | -0.111 | 0.021 | 0.039 | 0.103 |  |  |  |
| 5 | -0.226* | -0.041 | -0.081 | 1 | 0.164* | 0.275* | -0.081 | -0.088 |  |  |  |
| 6 | -0.112 | -0.026 | -0.111 | 0.164* | 1 | 0.126 | -0.021 | -0.049 |  |  |  |
| 7 | -0.142* | -0.150* | 0.021 | 0.275* | 0.126 | 1 | -0.325* | -0.177* |  |  |  |
| 8 | 0.091 | 0.059 | 0.039 | -0.081 | -0.021 | -0.325* | 1 | 0.584* | 0.050 | 0.147 | 0.082 |
| 9 | 0.012 | 0.078 | 0.103 | -0.088 | -0.049 | -0.177* | 0.584* | 1 | 0.143* | 0.070 | 0.042 |
| 10 |  |  |  |  |  |  | 0.050 | 0.143* | 1 | 0.037 | 0.109 |
| 12 |  |  |  |  |  |  | 0.147 | 0.070 | 0.037 | 1 | 0.078 |
| 13 |  |  |  |  |  |  | 0.082 | 0.042 | 0.109 | 0.078 | 1 |

*$P<0.05$; 1 = nurse age; 3 = nurse working years; 4 = marital status of nurse; 5 = night shift last month; 6 = weekly hours of working; 7 = mental health score; 8 = relational Care score; 9 = nursing care satisfaction score; 10 = patient age; 12 = annual income (Yuan) of patients; 13 = marital status of patient

**Table 3. Chi-square test and nonparametric test results for study variables ($\chi^2$ / Z value).**

|    | 1     | 3     | 4     | 5      | 6      | 7     | 8      | 9       | 10     | 12     | 13     |
|----|-------|-------|-------|--------|--------|-------|--------|---------|--------|--------|--------|
| 2  | 0.073 | 8.168 | 2.041 | -0.049 | -0.059 | 2.261 | -0.223 | -1.731  |        |        |        |
| 11 |       |       |       |        |        |       | -0.615 | 11.636* | -0.246 | -1.019 | -0.132 |

*$P<0.05$; 1 = nurse age; 2 = nurse gender; 3 = nurse working years; 4 = marital status of nurse; 5 = night shift last month; 6 = weekly hours of working; 7 = mental health score; 8 = relational Care score; 9 = nursing care satisfaction score; 10 = patient age; 11 = patient gender; 12 = annual income (Yuan) of patients; 13 = marital status of patient

-0.325, $p < 0.05$) and nursing care satisfaction score (r = -0.177, $p < 0.05$). Also, relational care score, patients age and gender were correlated with nursing care satisfaction score (r = 0.584 and r = 0.143, $x^2$ = 11.636, $p < 0.05$).

Descriptive information of nurses and patients, mental health, relational care score and nursing care satisfaction score were entered into the model. The final structural equation model fitted well with the study data ($\chi^2$ = 439.188, $p < 0.001$, GFI = 0.959, AGFI = 0.939, RMSEA = 0.049) (Fig 1). All effects were significant ($p < 0.001$).

Table 4 shows the standardized direct, indirect, and total effects, and the 95% confidence interval (CI) for each construct. Four direct, one indirect, and five total effects were significant. Descriptive information of nurses had a direct impact on nurses' mental health (direct

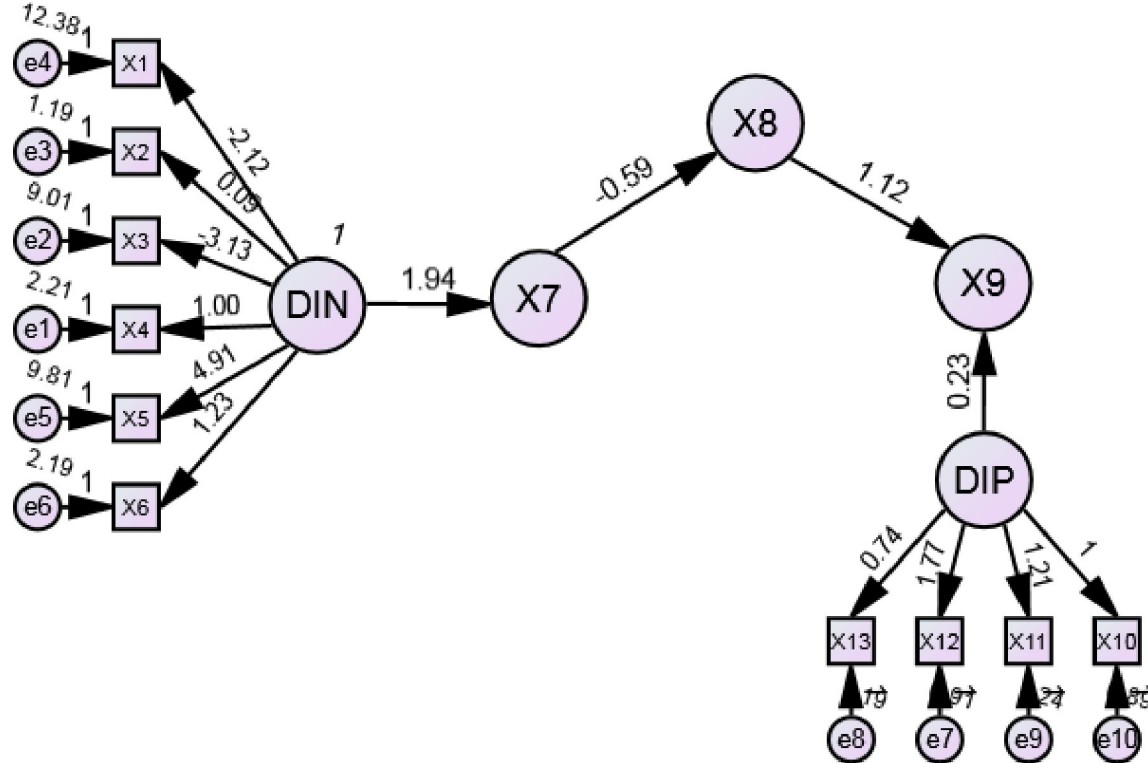

**Fig 1. Modified model of nurse's mental health and relational care score and nursing care satisfaction data in patients.** *The goodness-of-fit indices were: CFI = 0.962, NFI = 0.951, TLI = 0.949, GFI = 0.959, AGFI = 0.939, RMSEA = 0.049. *$P < 0.05$; X1 = nurse age; X2 = nurse gender; X3 = nurse working years; X4 = marital status of nurse; X5 = night shift last month; X6 = weekly hours of working; DIN = descriptive information of nurses; X7 = mental health score; X8 = relational Care score; X9 = nursing care satisfaction score; DIP = descriptive information of patients; X10 = patient age; X11 = patient gender; X12 = annual income (Yuan) of patients; X13 = marital status of patient.

**Table 4. Standardized direct, indirect, and total effects of variables in the final model (95% CI).**

| Exogenous variable | Endogenous variable | Direct effect | Indirect effect | Total effect |
|---|---|---|---|---|
| Nurses descriptive information | Mental health | 0.612 (0.419–0.713)* | | 0.612 (0.419–0.713)* |
| | Relational Care score | | 0.114 (0.129–0.201) | 0.114 (0.129–0.201) |
| | Nursing care satisfaction score | | 0.056 (0.023–0.051) | 0.056 (0.023–0.051) |
| Mental health | Relational Care score | 0.493 (0.298–0.428)* | | 0.493 (0.298–0.428)* |
| | Nursing care satisfaction score | | 0.051 (0.032–0.074)* | 0.051 (0.032–0.074)* |
| Relational Care score | Nursing care satisfaction score | 0.232 (0.057–0.172)* | | 0.232 (0.057–0.172)* |
| Patients descriptive information | Nursing care satisfaction score | 0.057 (0.347–0.493)* | | 0.057 (0.347–0.493)* |

effect = 0.612, 0.419–0.713). Nurses' mental health had a direct effect on relational Care score (direct effect = 0.493, 0.298–0.428) and an indirect effect on nursing care satisfaction score (indirect effect = 0.051, 0.032–0.074). Relational Care score and patient's descriptive information had also a direct effect on nursing care satisfaction score (direct effect = 0.232, 0.057–0.172 and 0.057, 0.347–0.493).

## Discussion

Patient satisfaction and their expectations of care are important indicators of high-quality nursing care in hospitals and healthcare facilities. Because of this, these centers are looking for ways to improve patient satisfaction by providing good services. It is important to understand what factors affect how satisfied patients are.

The current study found that male patients were more satisfied with nursing care than female patients. This result is consistent with the findings of Rafii et al (2009) and Guo et al. (2023), which also showed that male patients were more satisfied with the nursing care they received [15, 24]. Men usually have lower expectations and are less sensitive than women, and they often receive more information from nurses, which might make them more satisfied with the nursing care [15, 25].

Another finding from this study is that age had a positive correlation with nursing care satisfaction; In addition, age had a direct effect on nursing care satisfaction. This means that patients who are older were more satisfied than younger patients. These results are consistent with other studies [15, 26, 27].

It appears that older patients have lower expectations for nursing services and fewer needs compared to younger patients, which makes them more likely to be satisfied. Additionally, older patients may be more socially aware, more forgiving, and more appreciative and empathetic towards their caregivers [27].

Nurses often interact the most with patients. The correlation analysis of this study showed that when nurses have good relationships with patients, the patients are more satisfied with the care they receive. Also, analysis indicated that the nurse-patient relational care score directly affects nursing care satisfaction, which is consistent with other studies results [15, 28]. Good communication between nurses and patients is important for the patients and their families to feel satisfied with the care. If communication is poor, it can cause a lack of trust and harm the relationship between the patient and the nurse [29, 30]. Some studies suggest that better training can help nurses improve their communication skills, which can lead to safer care, better health results, patients following their treatment plans, and overall satisfaction with the care they receive. Moreover, it helps healthcare workers feel more valued and effective [31, 32].

The role of the nurse in the relationship between nurse-patient relational care score with nursing care satisfaction has been less investigated. One of the aspects that can be paid

attention to is the role of the nurse's mental health in the relationship between nurse-patient relational care score with nursing care satisfaction.

However, this issue has not been given much attention in the study so far. In our study, we found that the mental health of nurses can indirectly affect patient satisfaction through influencing the nurse-patient relational care. Good mental health in nurses can increase their self-confidence and ability to do their job well by talking with patients. This helps them share information effectively and handle difficult situations without feeling stressed. Showing respect and understanding the needs of patients with dual diagnosis is very important in providing good care. Research has shown that understanding the perspectives of these individuals can be very beneficial [33].

Considering the importance of the role of mental health of nurses in the relationship between nurse-patient relational care score with nursing care satisfaction, it's important to focus on what affects nurses' mental health. In this study, it was found that older nurses who have been working longer tend to have better mental health, meaning they experience less stress, anxiety, and depression. This is likely because they have more experience and maturity. In fact, age and years of work were found to be factors that reduce the chances of poor mental health. Other studies have also shown that younger nurses with less experience are more likely to feel stressed [34, 35].

In this research, it was found that the more night shifts nurses worked each month, the worse their mental health tended to be. A study conducted by Torquati et al., (2019) showed that working shifts increased the chances of having mental health problems like depression and anxiety [36]. Normally, people sleep at night and are awake during the day. However, to keep the healthcare system running well, many healthcare workers, like nurses, have to work night shifts either regularly or permanently [37].

The night nurses give constant care and watch over patients, and they also give out medicine. However, working night shifts regularly can lead to poor mental health, which can cause them to feel different emotions, have trouble thinking, not want to do things, do their job worse, be more likely to get hurt, and have changes in their body [38]. Some studies showed that nurses who moved from night shifts to day shifts experienced a significant reduction in symptoms of depression and anxiety over a 2-year period [39, 40].

It is important to acknowledge some limitations of this study. Firstly, the use of a cross-sectional design prevents the identification of a causal relationship between variables. Secondly, the number of questions and the time required to fill them can make the participants tired and impatient.

## Conclusion

Overall, this study found that the mental health of nurses is one of the indirect factors affecting patients' satisfaction. So, focusing on nurses' mental well-being and helping them feel better could be a way to make patients more satisfied with their nursing care.

## Acknowledgments

We thank all the nurses and patients who participated in this study.

## Author Contributions

**Conceptualization:** Hui Huang, Haosheng Ni.

**Data curation:** Jing Cui, Hua Zhang, Haosheng Ni.

**Formal analysis:** Hui Huang, Jing Cui, Hua Zhang, Yuhui Gu, Haosheng Ni, Ya Meng.

**Investigation:** Hui Huang, Hua Zhang, Haosheng Ni, Ya Meng.

**Methodology:** Hui Huang, Jing Cui, Hua Zhang, Yuhui Gu, Haosheng Ni, Ya Meng.

**Project administration:** Jing Cui, Ya Meng.

**Resources:** Haosheng Ni, Ya Meng.

**Software:** Jing Cui.

**Supervision:** Hua Zhang, Yuhui Gu.

**Validation:** Hua Zhang, Yuhui Gu, Haosheng Ni, Ya Meng.

**Visualization:** Yuhui Gu, Ya Meng.

**Writing – original draft:** Ya Meng.

**Writing – review & editing:** Ya Meng.

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
