## [Decision Letter · Decision Letter 0]

19 Jul 2024

PONE-D-24-21218The role of mental health in the relationship between nursing care satisfaction with nurse-patient relational care in Chinese emergency department nursingPLOS ONE

Dear Dr. Meng,

Thank you for submitting your manuscript to PLOS ONE. After careful consideration, we feel that it has merit but does not fully meet PLOS ONE’s publication criteria as it currently stands. Therefore, we invite you to submit a revised version of the manuscript that addresses the points raised during the review process.

We look forward to receiving your revised manuscript.

Kind regards,

Sadia Malik, Ph.D.

Academic Editor

PLOS ONE

Journal Requirements:

2. Thank you for stating the following financial disclosure: "This study was funded by Nantong Science and Technology Project, China (Grant number: MS22022113)."

3. In the online submission form, you indicated that " The data used to support the findings of this study are available from the corresponding author upon request."

Reviewers' comments:

Reviewer's Responses to Questions

**Comments to the Author**

1. Is the manuscript technically sound, and do the data support the conclusions?

Reviewer #1: Partly

Reviewer #2: Yes

2. Has the statistical analysis been performed appropriately and rigorously? 

Reviewer #1: No

Reviewer #2: Yes

3. Have the authors made all data underlying the findings in their manuscript fully available?

Reviewer #1: Yes

Reviewer #2: Yes

4. Is the manuscript presented in an intelligible fashion and written in standard English?

Reviewer #1: No

Reviewer #2: Yes

5. Review Comments to the Author

Reviewer #1: ID: PONE-D-24-21218

Title: The role of mental health in the relationship between nursing care satisfaction with nurse-patient relational care in Chinese emergency department nursing

Thank you for providing a chance to review this manuscript.

Detailed information:

Keywords："mental health" should read "Mental health".

Abstract

Line 28 - 38, page 2: Pay attention to the norms of punctuation, with spaces before and after "=" and "<".

Overall: The abstract requires a brief overview of the background, purpose, methods, results, and conclusions of the article. What's your conclusion?

Introduction

Line 61 - 63, page 4: “hospitals and many healthcare centers strive to fulfill patients' needs through the provision of quality services” Is it possible to give some relevant examples so that readers can understand.

Line 76 - 77, page 4: “Emergency departments have a distinct advantage in influencing a patient's initial perception as they navigate through the healthcare system” How is the emergency department different from other departments and can it be explained?

Method

Study design and participants

Line 98 - 103, page 5: Are all tertiary hospitals in the area included?

Line 99, page 5: “November 10 to December 30, 2023.” When were the surveys of nurses and patients respectively conducted? Please be specific.

Line 102, page 5: “only Grade III hospitals were selected” Why choose more than just secondary hospitals? Why choose only tertiary hospitals? What are the advantages of tertiary hospitals over other levels of hospitals for this study?

Line 104, page 5: Why were emergency department nurses used as survey respondents? And not other unit nurses? How is the emergency department different from other departments?

Line 104 - 108, page 5: “with at least one year of working experienc”“nurses who had taken a leave for more than six months in the previous year due to different reasons” Why the one-year and six-month timeframes? Is there a basis for this? If so, please explain.

Line 129 - 130, page 7 - 8: 1) Pay attention to the norms of punctuation, please verify that punctuation is used correctly throughout the text. 2) Please use a 3-wire meter. 3) Personally, I don't think there is a great need to set up a table to account for this information, and it's fine to account for it directly in the body of the text.

Demographic information

Line 131 - 135, page 8: Is a demographic table available? If not, it is recommended that a demographic table be drawn up so that the demographic characteristics of nurses and patients are displayed more clearly.

Nursing care satisfaction scale

Line 138 - 139, page 8: The result of the application? Good or bad? Please elaborate.

Line 143, page 8: “Cronbach's α coefficient” Note that the full name is alpha.

Statistical analysis

Line 164 - 169, page 9: 1) chi-square test requires italics. Please correct.

Line 170 - 172, page 10: What is the recommended range to which these values apply? What is the literature on which they are based? Please provide details.

Results

Line 174, page 10: Are 532 nurses and 532 patients the final sample size? Were there any missing samples during the survey? How were the missing samples handled?

Overall: 1) Pay attention to the norms of punctuation, please verify that punctuation is used correctly throughout the text. 2) My suggestion is to divide it into subheadings according to the content of the research.

There are some problems in this article: 1) punctuation conforms to the use of incorrect; 2) much of the content is not detailed enough; 3) there are some logical errors.

Thank you and my best,

Your reviewer

Reviewer #2: This article is a good and high quality work. Also, Its topic is interesting for researchers. The methodology and method is sound. Study recommendations are useful. Hence in my opinion, this manuscript is eligible for publication.

My decision: Accept

6. PLOS authors have the option to publish the peer review history of their article (what does this mean?). If published, this will include your full peer review and any attached files.

Reviewer #1: No

Reviewer #2: **Yes: **Dr. Esmail Khodadadi

---

## [Author Response · Author response to Decision Letter 0]

31 Jul 2024

I thank you, dear reviewers. Your valuable comments can make me learn more and more. I tried my best to do the revisions according to your comments. 

Also, we tried to edit the entire text.

Reviewer comment: Keywords："mental health" should read "Mental health".

we corrected (line 41). 

Reviewer comment: Line 28 - 38, page 2: Pay attention to the norms of punctuation, with spaces before and after "=" and "<". 

According to one of the articles that was recently published in PLOS ONE journal (Waddington EE, Allison DJ, Calabrese EM, Pekos C, Lee A, et al. (2024) Orienteering combines vigorous-intensity exercise with navigation to improve human cognition and increase brain-derived neurotrophic factor. PLOS ONE 19(5): e0303785. https://doi.org/10.1371/journal.pone.0303785), I edited the punctuation in the entire text. 

Reviewer comment: Overall: The abstract requires a brief overview of the background, purpose, methods, results, and conclusions of the article. What's your conclusion?

we added conclusion and edited. 

Reviewer comment: Line 61 - 63, page 4: “hospitals and many healthcare centers strive to fulfill patients' needs through the provision of quality services” Is it possible to give some relevant examples so that readers can understand. 

We added (lines 46-50). 

Reviewer comment: Line 76 - 77, page 4: “Emergency departments have a distinct advantage in influencing a patient's initial perception as they navigate through the healthcare system” How is the emergency department different from other departments and can it be explained?

A visit to the emergency department is usually the first time a patient interacts with a hospital system. Thus, this is a special opportunity to make a positive first impression. 

We added these sentences (lines 65-67).

Reviewer comment: Line 98 - 103, page 5: Are all tertiary hospitals in the area included?

Yes, all Level-III hospitals were included in the study. All the cities of Hubei province except for three cities had this model of hospitals.

Reviewer comment: Line 99, page 5: “November 10 to December 30, 2023.” When were the surveys of nurses and patients respectively conducted? Please be specific.

We added (lines 112 to 115).

Reviewer comment: Line 102, page 5: “only Grade III hospitals were selected” Why choose more than just secondary hospitals? Why choose only tertiary hospitals? What are the advantages of tertiary hospitals over other levels of hospitals for this study?

We had to choose a hospital model to match the situation. Therefore, this question arose in each of the hospitals we chose. Therefore, choosing these types of hospitals in our research does not have a special advantage. In the future, research can be done according to the quality level of hospitals and comparison of services.

Reviewer comment: Line 104, page 5: Why were emergency department nurses used as survey respondents? And not other unit nurses? How is the emergency department different from other departments?

A visit to the emergency department is usually the first time a patient interacts with a hospital system. Thus, this is a special opportunity to make a positive first impression. Therefore, the nurses who work in these departments are among the first medical staff of the hospital who deal with the patient.

We added these sentences (lines 65-67).

Reviewer comment: Line 104 - 108, page 5: “with at least one year of working experienc”“nurses who had taken a leave for more than six months in the previous year due to different reasons” Why the one-year and six-month timeframes? Is there a basis for this? If so, please explain.

This time period was chosen according to a pattern taken from other articles.

Reviewer comment: Line 129 - 130, page 7 - 8: 1) Pay attention to the norms of punctuation, please verify that punctuation is used correctly throughout the text. 2) Please use a 3-wire meter. 3) Personally, I don't think there is a great need to set up a table to account for this information, and it's fine to account for it directly in the body of the text.

Yes, according to your comment, I removed the table and entered the content in the text (lines 116-118).

Line 131 - 135, page 8: Is a demographic table available? If not, it is recommended that a demographic table be drawn up so that the demographic characteristics of nurses and patients are displayed more clearly: 

I added Table 1.

Line 138 - 139, page 8: The result of the application? Good or bad? Please elaborate.

This questionnaire is not classified. In addition, we were looking for quantitative data to evaluate its relationship with mental health. 

Line 143, page 8: “Cronbach's α coefficient” Note that the full name is alpha.:

I corrected (lines 133, 141, 152). 

Line 164 - 169, page 9: 1) chi-square test requires italics. Please correct.

I corrected. 

Line 170 - 172, page 10: What is the recommended range to which these values apply? What is the literature on which they are based? Please provide details.

I added lines 162 to 164. 

Line 174, page 10: Are 532 nurses and 532 patients the final sample size? Were there any missing samples during the survey? How were the missing samples handled?

I added lines 166 to 170.

Pay attention to the norms of punctuation, please verify that punctuation is used correctly throughout the text. 

We edited it.

---

## [Editor Report · Decision Letter 1]

20 Aug 2024

The role of mental health in the relationship between nursing care satisfaction with nurse-patient relational care in Chinese emergency department nursing

PONE-D-24-21218R1

Dear Dr. Meng,

We’re pleased to inform you that your manuscript has been judged scientifically suitable for publication and will be formally accepted for publication once it meets all outstanding technical requirements.

Kind regards,

Sadia Malik, Ph.D.

Academic Editor

PLOS ONE
---

## [Editor Report · Acceptance letter]

22 Aug 2024

PONE-D-24-21218R1 

PLOS ONE

Dear Dr. Meng, 

I'm pleased to inform you that your manuscript has been deemed suitable for publication in PLOS ONE. Congratulations! Your manuscript is now being handed over to our production team.

Kind regards, 

on behalf of

Dr. Sadia Malik 

Academic Editor

PLOS ONE